# Is Transient and Persistent Poverty Harmful to Multimorbidity?: Model Testing Algorithms

**DOI:** 10.3390/ijerph16132395

**Published:** 2019-07-05

**Authors:** Sukyong Seo, Young Dae Kwon, Ki-Bong Yoo, Yejin Lee, Jin-Won Noh

**Affiliations:** 1College of Nursing, Eulji University, Seongnam 13135, Korea; 2Department of Humanities and Social Medicine, College of Medicine and Catholic Institute for Healthcare Management, The Catholic University of Korea, Seoul 06591, Korea; 3Department of Health Administration, Department of Information & Statistics, Yonsei University, Wonju 26493, Korea; 4Department of Healthcare Management, Eulji University, Seongnam 13135, Korea; 5Global Health Unit, Department of Health Sciences, University Medical Centre Groningen, University of Groningen, 9713 GZ Groningen, The Netherlands

**Keywords:** multimorbidity, poverty, Korean Health Panel, model testing algorithm, dynamic

## Abstract

Multimorbidity, the coexistence of two or more long-term medical conditions in one person, has been known to disproportionally affect the low-income population. Little is known about whether long-term income is more crucial for multimorbidity than income measured in one time point; whether persistent poverty is more harmful than transient one; how changes in wealth affect multimorbidity. This is a longitudinal study on a population representative dataset, the Korean Health Panel (KHP) survey (2010–2015). A multivariate analysis was conducted using logistic regressions. A variety of income and wealth variables was investigated. Low-income Koreans (lowest 20%) were more likely to have multiple disorders; average income was more significantly associated with multimorbidity than the yearly income measured for the same year; persistent episodes of poverty had a greater hazard than transient ones; and income changes appeared to be statistically insignificant. We found that long-term income and persistent poverty are important factors of multimorbidity. These findings support the importance of policies reducing the risk of persistent poverty. Policies to promote public investment in education and create jobs may be appropriate to address multimorbidity.

## 1. Introduction

Socioeconomically deprived individuals experience a greater burden of multimorbidity and are affected at an earlier age [1,2,3]. In 2017, Katikireddi et al. reported that the risk of developing multimorbidity was approximately 1.5 times higher in poor than in rich individuals [3]. We hypothesized that steeper coefficients for income appeared when analyzing the recent data in South Korea because of the two contextual health system factors: 1) social health insurance with low depth of coverage level and market-based healthcare delivery and 2) limited primary healthcare. Although the country has a universal health insurance, the depth of coverage level tends to be low. Out-of-pocket consumer payments are relatively large, ranging from 20% to 55% for covered services, and many expensive and essential services are not covered [4]. Therefore, those who are poor have a greater disease burden than the affluent. The more pressing issue related to the management of multimorbidity is the limited coordination in primary healthcare [5]. Since compensated on a fee-for-service (FFS) basis, physicians are likely to see more patients rather than spending time counseling and teaching self-management. Patients with multimorbidity have a combination of physical, psychological, and social problems, and consequently need time, empathy, and a holistic patient-centered approach to care [6]. However, they receive too little time and empathy in South Korea. We believe that this has a detrimental effect on the underachievement in healthcare for the poor, which motivates us to investigate the association between poverty and multimorbidity in South Korea.

Previous studies have shown the importance of income in multimorbidity development [1,2,3]. Although income and multimorbidity development were strongly, negatively correlated in these studies, an income level measured at one time point may be limited to reflect material resources available to an individual [7]. Moreover, it may be unsuitable to link income level and health status measured for the same year in case of chronic diseases. Researchers in the field of health economics and public health showed that income changes, over and above income levels, play an important role in determining an individual’s health status. People with decreased income level over time have poorer health outcomes when compared to those with stable or increased income levels [8]. These researchers also found that poverty is dynamic not fixed: some people face long lasting financial hardship, while others move in and out of poverty in various ways at different time periods. 

Evidence of focus on the dynamic aspects of poverty is limited among researches on multimorbidity. Therefore, we aimed to investigate the association between income status and multimorbidity by incorporating several income measures, such as income levels, income changes, and poverty experiences. Research questions included “Is long-term (permanent) wealth measure more important for multimorbidity than income measured at one time point?”, “Is sustained financial hardship more harmful than transient poverty experience?”, and “What is the effect of changes in income on multimorbidity?”

## 2. Materials and Methods 

### 2.1. Data and Subjects

This study is a longitudinal analysis on a population representative dataset, the Korean Health Panel (KHP) survey (2010–2015). The survey commenced in 2008 and has been followed up annually by the Korea Institute for Health and Social Affairs (KIHASA) and the Korea National Health Insurance Service. Based on the Population and Housing Census in 2005, the initial KHP sample (n= approximately 23,000 in 2008) was designed as a nationally representative cohort of non-institutionalized men and women living in South Korea across all age groups. The stratified sampling strategy has been previously described [9]. Strenuous efforts were made to follow up panel subjects over time. Attrition has occurred among specific groups of the population, including non-homeowners, low income households, younger population, and the highly mobile. Thus, longitudinal weights were analyzed and calculated to control these biases. For this study’s analysis, a balanced panel of 12,155 individuals for six waves (2010–2015) was established.

The survey data were collected through face-to-face interviews using self-report questionnaires that include demographic information, changes in household members, household income levels, health behaviors, and usage and expenditure of healthcare services. Details on chronic diseases were available in separate files. Trained interviewers collected the data, and participants had to provide official records of physician visits and details regarding pharmacy usage for long-term medical conditions. The data were extracted from KIHASA under a special permission for research purposes [10]. Ethical approval for this data was obtained by KIHASA. The procedures of this study were reviewed and approved by the Institutional Review Board of the Eulji University of Korea (EUIRB2018-67).

### 2.2. Variables and Measurement

#### 2.2.1. Multimorbidity Development

We defined 46 chronic conditions based on the ICD-10 codes from Bussche et al. [11]. The outcome of interest, multimorbidity, was defined as an individual having three or more chronic conditions out of the list in 2015. The criterion of the three conditions was considered to be more valid cut-off for multimorbidity in patients treated in the ambulatory care setting, instead of the usual criterion of the two chronic conditions [11].

#### 2.2.2. Income and Poverty Dynamics

The KHP survey collects information on total household incomes from all sources—employment, pension, and financial investment—for all house members. A derived variable with five quintiles of household income adjusted by family size was provided in the original dataset and employed in our analysis. With the numeric value of the total household income, two adjustments were performed. First, the annual household income was divided by the square root of the number of household members in the current year followed by OECD publications [12]. The needs of a housing space and electricity will not be two times as high for two household members than for a single member. Therefore, this study used a square root scale to equalize the income across the household accounting for number of household members. Second, since we used data over six years (2010–2015), the income values are adjusted by the retail price index (January 2015 = 100) [13]. Using the adjusted total household earning, we generated three income dynamic measures: income levels, poverty experiences, and income changes. 

First, yearly income variables were calculated for each year during study period (2010–2015). Next, the five-year (2010–2014) average was also calculated, and the association between income levels and health status in the same year was determined [8]. Since we defined the outcome measure as whether an individual had three or more chronic conditions in 2015, the time when our independent variable, income, was measured was ahead of the year of 2015. 

Second, we derived a set of variables assessing people’s poverty experiences over five years by measuring the duration of poverty following a previous landmark study [8]. Poverty experience 1 is the number of years were that the household income was below half of the average for each year (calculated from the original data file with all observations including household income newly added to the sample each year). Poverty experience 2 is the number of years that the income was in 20 percentile of the distribution. A categorical variable indicating the stability of poverty experiences; Category 1 as more than 3 years in 0–40 percentile and no years in 61–100 percentile, category 2 as 1–2 years in 0–49 percentile and no years in 61–100 percentile, category 3 as income in 41–59 percentile or a combination of more years in the bottom 40 percentile and no years in the top 40 percentile, category 4 as 1–2 years in the top 40 percentile and no years in the bottom 40 percentile, category 5 as more than 3 years in the top 40 percentile and no years in the bottom 40 income percentile.

The final indicators measured the degree of income changes for six years of our analytical data. The simple monetary difference was between the income level in 2015 and 2010. A categorical variable identified households with large increases and decreases in income (>30%) from 2010 to 2015. Volatility was measured by the standard deviation of each household income during study period (2010–2015).

#### 2.2.3. Control Variables

There are a number of confounders such as educational and marital status, employment, lifestyle factors, and autonomy at work regarding the association between income and multimorbidity. Simple models were constructed in our study and controlled for age, sex, and homeownership only. Although the effect of confounders should be obviously controlled, this study seeks to carefully explore the direct relationship between income and multimorbidity. Therefore, only the straightforward confounders were included.

### 2.3. Statistical Analysis

Since it may be unsuitable to link income and health in the same year [8], we introduced health measures that preceded income measures. For example, we examined the association between the income levels from 2010 to 2014 and multimorbidity development in 2015. To introduce a time dimension, previous health status (measured as the number of chronic diseases in the previous year) and income over times have been added. 

In order to examine the relation of multimorbidity to income, a multivariate analysis was conducted using logistic regressions. Performing both logistic and probit, we found that the logistic model had very similar values for AIC/BIC to the probit (logistic models had a bit smaller values). The significance of the models was assessed using the Wald F-statistic.

First, a crude model was devised with age, sex, and initial health status as control variables to investigate whether long-term income was more crucial than income data collected at one setting; we added income measured at four time points—current year (t), previous year (t-1), initial year (t-6), and the average over the first five years (t-6 through t-1). The relative importance for multimorbidity of each income measure was assessed by comparing the joint significance of each set of quintiles and pattern of ORs. Next, we investigated whether persistent poverty was associated with multimorbidity than the transient one. Three poverty duration variables were added. Last, the effect of income changes on multimorbidity was also investigated. Three measures of income changes were added to a crude model. The significance of each income change measure was assessed using Wald F-statistics to compare which appeared more important. The goodness of fit test was performed to compare which model appeared confident. We performed our logistic analysis to capture any unknown confounding factor affecting every family member in a household simultaneously (using a stata command, vce). All the analyses were performed using STATA 13.1 (StataCorp, College Station, TX, USA).

### 2.4. Research Ethics

All subjects gave their informed consent for inclusion before they participated in the study. The study was conducted in accordance with the Declaration of Helsinki, and the protocol was approved by the Ethics Committee of Eulji Institutional Review Board (EUIRB2018-67).

## 3. Results

Table 1 shows a summary of the statistics on the study population. Overall, 12,250 observations were used in the study. Approximately 33% of the sample had multimorbidity (three or more chronic diseases at the same time). The income dynamics was quite considerable over the six-year period. For instance, 45% of the sample experienced poverty (living with less than average income) in a study period.

Table 2 presents the cross-sectional relationship between income and multimorbidity by adding the previous health status and then income indicators over time. Column 1 indicates the OR of current income. People in the bottom 20% of the income distribution (1st quintile) are approximately 7.6 times more likely to have three or more chronic diseases at the same time. Column 2 shows the effects of adding initial health to column 1logistic regression model. The OR for initial health is large (OR = 3.50, *p* < 0.001). Adding initial health results substantially increased the overall significance of the models (test of goodness of fit: 3804→75,472). ORs for current income were reduced (by 48.8%, 31.2%, 10.1%, and 2.0%) but remained statistically significant. It means that selection effects explain a small proportion of the cross-sectional relationship between income and multimorbidity, the main direction of causation comes from income and multimorbidity. The last column presents the cross-sectional relationship between current income and multimorbidity by adding the five-year average income, after adjusting for covariates. The five-year average value was not statistically significant.

Table 3 shows the relative contribution of income measured at different time points: current year (2015), previous year (2014), initial year (2010), and five-year average. The coefficients in the first column were consistent with those in column 2 at Table 2 and re-presented to compare the income indicators. The most significant and steepest association was with the five-year average income.

Table 4 explores the effects of poverty experience on multimorbidity, showing that duration of poverty was significantly associated with multimorbidity between 2010 and 2014. All four models have the same dependent variable and covariates as the models in Table 3 and different income measured. People who experienced persistent poverty for four or five years had the worst health status. The risk of multimorbidity consistently reduced as the poverty duration decreased. The poverty measure among people at the bottom 20% appeared to be more strongly associated with multimorbidity than those who have less than half the average income.

We investigated the effects of poverty stability on multimorbidity and the relationship between income changes and multimorbidity. Initial income appeared to be statistically significantly associated with multimorbidity. None of the income change measures were significant and presented the result in Appendix A.

## 4. Discussion

This study aimed to investigate whether long-term income is more crucial for multimorbidity than income measured cross-sectionally and to measure the effect of changes in income on multimorbidity. Our analysis of the KHP showed that the average income was more significantly associated with multimorbidity than current income; those who experience persistent poverty are more at risk than transient episodes of poverty; and income change appeared to be statistically insignificant. This result implies that long-term wealth and persistent poverty are crucial for accounting for multimorbidity development. Previously Benzeval and Judge (2001) examined the British Panel Data and found that people with decreased income level over time had poorer health outcomes when compared to those with stable income [8] dray. Although they did not used health outcomes related multimorbidity, their findings motivate us to investigate the association between dynamic aspects of poverty and multimorbidity development.

Income change was insignificantly associated with multimorbidity in the current analysis. For men, income quintiles were not significant (see Appendix A). These results are not unexpected. Many previous studies have shown that people with decreasing income levels over time have poorer health outcomes than those with stable or increasing income. However, some studies also reported little effects of income change on health status. For working class women and all men, loss of income was weakly associated with effects on their health status [14,15]. This may be due to relatively short panels that our analysis is based on. People can maintain their living standards by paying from their saving rather than changing their immediate standards of living. Further investigations with longer runs of panel data are required to examine the effects of the dynamics of wealth change in more detail. 

The negative association between current income and multimorbidity is not surprising and is consistent with the results of several international studies on the association between income status and multimorbidity [1,6]. However, the association in this study is much weaker than the previous cross-sectional studies (ORs for current income in Table 2 range from 1.15 to 1.50). For example, Barnett et al. found that Scottish people in the bottom 20% of income distribution were approximately 2.1 times more likely to have multimorbidity than those at the top income group [1]. However, the weak association in this study may also be consistent with other studies using longitudinal data [3,16,17], with the lowest and highest SES (Socioeconomic status) OR of 1.20 to 1.91 to compare the results. 

The weaker socioeconomic association in our data may be a result of differential reporting. As the Scottish study employed clinical data collected by GPs, our data on morbidities are based on self-reported information and included a list of both physical and mental health problems. Since self-report relies on an individual’s memory, the information may be inaccurate compared to clinical data and cannot be counted as a reliable source of his or her morbidities. Moreover, individuals with mental health problems are likely to under-report their health problems [18]. In addition, chronic conditions include some diseases associated with SES (e.g., chronic obstructive pulmonary disease and asthma), but others are not (e.g., pain, migraines), which are more highly associated with old age than income status. 

This study has some limitations. This study measured income not wealth. Therefore, the result couldn’t illuminate how poverty leads to multimorbidity. Poverty (whether income or wealth) would only jointly determine health in a community with poor social capital. There exist considerable regional differences of social infrastructure in Korea (in particular, between Seoul, the capital city, and cities located at southern part of Korea). People living in Seoul and those living in a southern city likely differ in their odds of multimorbidity. Although we tried to perform this study by employing a residential region dummy as a model specification, we should run separate analyses. Aggregating data and running one model for rich region and poor region would introduce place bias.

Another limitation of this study is lack of measures indicating quality of life/material well-being, access to care, and quality of care due to the data limitation. Diseases often have a combination of physical, psychological, and social problems, and consequently, which has a detrimental effect on the poor. These motivate us to investigate the association between poverty and multimorbidity. Although we tried to capture a longer term income and region effect, our results appeared no significant change. 

Further study is required to examine whether the weaker relation between income and multimorbidity remains in other subgroups. We also need to consider such influence as residential adverse exposures, household material well-being, and access to quality care. 

## 5. Conclusions

This study showed that poverty influenced the development of multimorbidity in South Korea. We found that long-term wealth and persistent poverty are crucial for multimorbidity. These findings support the importance of policies reducing the risk of persistent poverty than transient one. The Korean government has implemented a growth strategy which creates lots of jobs for a large workforce and heavy public investment in the education system to reduce the incidence of poverty [19]. This approach may be an appropriate response to address multimorbidity by promoting education and creating work opportunities for people with low income. 

We also recommend the recent governmental effort to introduce the Chronic Disease Management System to focus more on multimorbidity (it currently limits to management of hypertension and diabetes itself by making primary health care sector and public health centers participated). Policymakers should understand that people with a chronic disease often suffer from long term co-morbid conditions, which considerably affects their quality of life. The increase in multimorbid people would be another focus in terms of long-term growth in health care expenditures and workforce productivity. 

## Figures and Tables

**Table 1 ijerph-16-02395-t001:** Summary statistics on Korea Health Panel (2010–2015).

(*N* = 12,250)		
Variables		*n* (%) / mean (SD)
Number of chronic conditions in 2015	0	7145 (58.33)
	1-2	3135 (25.59)
	3+	4010 (32.73)
Age (at year 2015)		45.7 (22.1)
Sex	male	5879 (47.99)
	female	6371 (52.01)
Household income quintiles at year 2015	1 (poorest)	1858 (15.17)
	2	2419 (19.75)
	3	2517 (20.55)
	4	2850 (23.27)
	5 (richest)	2606 (21.27)
Five-year average income in 2015 term (1000 Korean won)		2216.4 (1292.6)
Poverty experience 1: Number of years with less than average income	0 year	9912 (54.67)
	1 year	3125 (17.24)
	2–3 years	1876 (10.35)
	4–5 years	3217 (17.74)
Poverty experience 2: Number of years with bottom fifth of income quintiles	0 year	9160 (74.78)
	1 year	1117 (9.12)
	2–3 years	889 (7.26)
	4–5 years	1084 (8.85)
Poverty duration	3+ years in bottom 2 quintiles and none in top 2 quintiles	3529 (28.81)
	1–2 years in bottom 2 quintiles and none in top 2 quintiles	713 (5.82)
	the rest	2376 (19.40)
	1–2 years in top 2 quintiles and none in bottom 2 quintiles	846 (6.91)
	3+ years in top 2 quintiles and none in bottom 2 quintiles	4786 (39.07)
Income changes	changes between year 2010 and year 2015	1100.2 (1,374.2)
	volatility (standard deviation of income 2010–2015)	671.3 (806.1)
Dummy variables for large income changes	30% increase (2010–2015)	5861 (47.84)
	30% decrease (2010–2015)	1549 (12.64)
	stable	4840 (39.51)
Home owners	Homeowners for 2010	8395(68.53)
	Rent (2 year + monthly + others) for 2010	3855(31.47)
ln (house value)	Self-reported house values for 2010	6.36(4.38)
ln (increase in house values)	Five-year average change in house values from previous year	3.36(3.45)

SD, standard deviation.

**Table 2 ijerph-16-02395-t002:** Effect of a time dimension on the cross-sectional relation between current income and multimorbidity.

Current Income (year 2015)	Odds Ratios for Multimorbidityª		
	Current income	+ health status (2010)	+Average income
1 (poorest)	7.93	3.72	3.66
2	2.90	1.88	1.86
3	1.33	1.15(ns)	1.14(ns)
4	0.99(ns)	0.97(ns)	0.95(ns)
5 (richest)	1.00	1.00	1.00
Initial health	-	3.50	3.50
Average income (2010–2015)	-	-	0.99(ns)
Wald *X*^2^ for the current income	713.23	241.16	171.73
Pseudo *R*^2^	0.1170	0.4524	0.4523
Wald *X*^2^ of error term	1102.16	2352.14	2351.35
Test of goodness of fit (Pearson *X*^2^)	3804.34	75472.81	84612.58

ª Model includes age, age squared, sex, and a set of proxy of house value (homeowner/rent, log of house value, log of average increase in house values). ORs are significant at p<0.10 unless indicated. (ns) indicates non-significant.

**Table 3 ijerph-16-02395-t003:** Income at four different time points and multimorbidity.

Income Quintile	Odds Ratios for Multimorbidity ª			
	Current income (*t*)	Income in previous year (*t*-1)	Initial income (*t*-6)	Five-year average income
1 (poorest)	3.79	3.30	2.54	2.77
2	1.90	1.80	1.58	1.54
3	1.16(ns)	1.23	1.26	1.28
4	0.97(ns)	1.02(ns)	0.93(ns)	1.08(ns)
5 (richest)	1.00	1.00	1.00	1.00
Wald *X*^2^ for the current income	241.16	178.59	106.78	136.43
Pseudo *R*^2^	0.4523	0.4474	0.4425	0.4445
Wald *X*^2^ of error term	2351.72	2304.85	2256.27	2271.47
Test of goodness of fit (Pearson *X*^2^)	75472.81	42436.10	47784.18	64452.99

ª Model includes age, age squared, sex, a set of proxy of house value, and initial health. ORs are significant at p<0.10 unless indicated. (ns) indicates non-significant.

**Table 4 ijerph-16-02395-t004:** Poverty duration and multimorbidity.

Number of Years Experienced Poverty	Odds Ratios for Multimorbidity ª	
	Less than Half of Average Income	Bottom fifth of Income Distribution
4 or 5 years	2.39	3.47
2 or 3 years	1.66	1.97
1 year	1.42	1.71
None (ref.)	-	-
*Wald F* test for poverty measures	126.40	163.73
Test of goodness of fit (Pearson *X*^2^)	59615.08	46455.66

ª Model includes age, age squared, sex, a set of proxy of house value, and initial health. ORs are significant at *p* < 0.10 unless indicated.

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
