# Peer review of "Is Transient and Persistent Poverty Harmful to Multimorbidity?: Model Testing Algorithms"

_ijerph, 2019, doi:10.3390/ijerph16132395_

Round 1

Reviewer 1 Report

This interesting study is both policy-timely and well-motivated; namely, the role of persistent poverty and long v/s short term resources on the development of co-morbid (long-term) conditions in the Korean population (KHP data for years 2010-2015). The authors proposed and estimated logistic regression model. The research design is reasonable and the data-set used has sufficient integrity. The empirical findings flow logically from the analysis. I have few but important concerns, though.

(1) Since men and women likely differ in their odds of developing multiple chronic conditions the authors SHOULD run separate logistic regression models (48% of the sample observations is Male and 52% Female. So, there are sufficient data points to run separate Male, Female models). Aggregating data and running one model for both males and females would introduce gender bias. Controls including a sex dummy is inadequate because of potential slope effects that are not captured. (2) I would like to see the authors use a proxy, such as Median residential House values, to capture long-term or permanent income effects (see, Lubiani, Okunade and Chen, 2018 paper in Atlantic Economic Journal. See, also Okunade, Suraratdecha and Benson paper in Health Economics journal, 2010). (3) Don't you think older population segment is likely to suffer more from multiple chronic co-morbid conditions? Please perform further data analysis and discuss this further. (4) The authors should also report some robustness tests to strengthen reader confidence in their results.  (5) Why pre-select logistic regression? Why not a probit model? Preference of one over another should be based on the empirical data distribution of the data being modeled. (6) Consider bringing in relevant Korean government policies in place or being considered to address long term co-morbid conditions and the implications for long term growth in health care expenditures, workforce productivity effects and quality of life improvements. I shall be happy to review a revised version, should the Editors invite you. 

Author Response

Editorial office review on duplication

Answer: Editorial office detected a high duplication rate coming from Benzeval & Judge (2001)’s study. Since we derived a set of variables assessing people’s poverty experiences and income changes following their study, the finding was not unexpected. We re-stated and deleted several sentences similar to Benzeval’s.

Reviewer 1

(1)     Since men and women likely differ in their odds of developing multiple chronic conditions the authors SHOULD run separate logistic regression models (48% of the sample observations is Male and 52% Female. So, there are sufficient data points to run separate Male, Female models). Aggregating data and running one model for both males and females would introduce gender bias. Controls including a sex dummy is inadequate because of potential slope effects that are not captured.

Answer: Following your suggestions, we ran a subset analysis for men and women separately. The result is presented in Table A in Appendix. Found that men are significantly different from women. Among men, in particular, most income measures have no effect on multimorbidity development. This is consistent with some previous studies. For example, Elder and Liker (1982) showed that for men (also working class women), loss of income was weakly associated with effects on their health status.

(2)     Don't you think older population segment is likely to suffer more from multiple chronic co-morbid conditions? Please perform further data analysis and discuss this further.

Answer: I agreed that the elderly may be different from the younger adults in terms of multimorbidity. Performed a subset analysis by age group and income group, we put its graphical presentation in Figure 1 at Appendix. Below is the result.

“…People with lower income were more likely to have multimorbidity than those with the highest income for all ages, except for those 35 years of age and under and at least 75 years of age. At 55 years of age, approximately 34% of population in the most affluent groups had multimorbidity. However, at 45 years of age, two lower income groups had similar rates of multimorbidity (30%) as the most affluent people aged 55; the economically disadvantaged people were affected 10 years earlier. By the age of 55, the poor were approximately 1.7 times more likely to have multimorbidity than the richest participants were. This was the greatest discrepancy in the proportion of patients with multimorbidity between income groups for all ages…”

(3)     I would like to see the authors use a proxy, such as Median residential House values, to capture long-term or permanent income effects (see, Lubiani, Okunade and Chen, 2018 paper in Atlantic Economic Journal. See, also Okunade, Suraratdecha and Benson paper in Health Economics journal, 2010).

Answer: We employed a set of proxy variables (house value, increase in house price, homeowners (yes/no), and residential region dummy) into our regression models. Found that two of the proxies were statistically significant. So, we re-performed regressions for all the models in Table 2 thru Table 4 to control for the long-term income effect. Highlighted in red is what appeared to change by adding up the proxy.

(4)     The authors should also report some robustness tests to strengthen reader confidence in their results. 

Answer: We performed and compared the Wald X2 test of independent errors to report which model appeared more confident. It is presented at the bottom of every table on the regression. Also presented and compared gof results.

Thanks to your suggestion, we decided to perform again our logistic analysis (for almost every model in Table 2-4) to capture any unknown confounding factor affecting every family member in a household simultaneously (using a stata command, vce(cluster). Found that many values of the standard errors had changed and updated our tables following the result.

(5)     Why pre-select logistic regression? Why not a probit model? Preference of one over another should be based on the empirical data distribution of the data being modeled.

Answer: Totally agreed. Performing both logistic and probit, we found that the logistic model had very similar values for AIC/BIC to the probit (the logit had a bit smaller values). See an example of one of our model from Table 2 as below. Also calculating predictions and comparing a probability scale (p, say), we found that the predictions were the same (see its scatter plot as below). We put down this into “2.3 statistical analysis” session.

Estimates from logit and probit model (with covariate as sex, age, age squared, and income quintiles)

Model

N

Df

AIC

BIC

Logit

12,250

8

8852.616

8911.923

Probit

12,250

8

8934.737

8994.043

Graph: logit prediction versus probit prediction

(6) Consider bringing in relevant Korean government policies in place or being considered to address long term co-morbid conditions and the implications for long term growth in health care expenditures, workforce productivity effects and quality of life improvements. I shall be happy to review a revised version, should the Editors invite you.

Answer: Thank you for your comment. We revised conclusion section with Korean government policies, as below.

“…The Korean government is implementing a growth strategy which creates lots of jobs for a large workforce and heavy public investment in the education system to reduce the incidence of poverty [19]…”

Reviewer 2 Report

I think this is an interesting and important contribution to the exploration of how income inequality influences health outcomes. The findings around the importance of poverty and persistent poverty in relation to multi-morbidity are fascinating and I really appreciate your attention to the health care system in Korea as a contextual factor.

The English is great and still needs a little editing....there are several sentence that use a plural noun inappropriately (e.g. "evidences" rather than "evidence") and a few places where the verb appears missing from a sentence.

While overall, I think the paper's methods and explanations of methods are excellent, there is room for some further clarification.

1) why did you divide income by the square root of family size?

2) did you consider creating a wealth/assets measure since it seems data are available or are you not able to create this measure.

3) Please be a bit clearer about what is in table 2----is the 2015 income used to predict multimorbidity ---the methods talk about using income two years prior to explain 2015 health status but table 2 seems to be cross-sectional....

4) Please clarify the text regarding Table 4....I read this as being about the same dependent variable and different measures of income...but text seems to imply that you are also examining multimorbidity over time

5) there are too many tables--- 3 or 4 would be ideal...Tables 5-6 could be relegated to an appendix and discussed in text...table 6 with non-significant findings seems unnecessary

Finally and most importantly, I believe the conclusions section could be strengthened by discussing study limitations.....most notably I would discuss the absence of place-related and other quality of life/material well-being indicators or access to care/quality of care indicators. The study does not illuminate how/why poverty leads to multimorbidity. The possibility exists that it is not poverty per se but rather how policy influences residential adverse exposures, household material well-being, and access to quality care (as suggested in the introduction) and thus the exclusion of these other indicators results in strong impacts of poverty rather than the more proximal determinants of health. Similarly, there should be a discussion of income vs wealth in this context. 

Author Response

Reviewer 2

(1)     why did you divide income by the square root of family size?

Answer: The needs of a household grow with each additional member but – due to economies of scale in consumption– not in a proportional way. Needs for housing space, electricity, etc. will not be three times as high for a household with three members than for a single person. Therefore, this study used a square root scale to equalize the income across the household accounting for number of household members. Recent OECD publications (e.g. OECD 2011, OECD 2008) comparing income inequality and poverty across countries use a scale which divides household income by the square root of household size.

(2)     did you consider creating a wealth/assets measure since it seems data are available or are you not able to create this measure.

Answer: We employed a set of proxy variables (house value, increase in house price, homeowners (yes/no), and residential region dummy) into our regression models to capture the effect of wealth. Found that two of the proxies were statistically significant. So, we re-performed regressions for all the models in Table 2 thru Table 4 to control for the long-term income effect. Highlighted in red is what appeared to change by adding up the proxy.

(3)     Please be a bit clearer about what is in table 2----is the 2015 income used to predict multimorbidity ---the methods talk about using income two years prior to explain 2015 health status but table 2 seems to be cross-sectional....

Answer: In Table2, we employed health measures that preceded income measures to introduce a time dimension to cross-sectional association. Then we moved into Table 3 to include income over time. We re-stated the method to make it clearly readable as below.

“…Since it may be inappropriate to link income and health status in the same year [8], we therefore employed health measures that preceded income measures. For example, we investigated the association between the average income from 2010 to 2014 and multimorbidity development in 2015. However, we did not believe the temporal orders matter when investigating income changes over time. The income data during the six-year study period were used when calculating income changes. To introduce a time dimension to conventional cross-sectional association between income and multimorbidity, previous health status (measured as the number of chronic diseases in the previous year) and income over time have been added. The odds ratio (OR) for current income was compared to better understand the effects of previous income and health on the cross-sectional association

(4)     Please clarify the text regarding Table 4....I read this as being about the same dependent variable and different measures of income...but text seems to imply that you are also examining multimorbidity over time

Answer: As you mentioned, I meant that all the models in Table 4 had the same dependent variable and different measures of income. I re-stated the method to make it clearer. Below is our new statement about it.

“…Table 4 explores the effects of poverty experience on multimorbidity, showing that duration of poverty was significantly associated with multimorbidity between 2010 and 2014. All four models have the same dependent variable and covariates as the models in Table 3 and different income measured. People who experienced persistent poverty for four or five years had the worst health status. The risk of multimorbidity was consistently reduced as the duration of poverty decreased. The poverty measure among people at the bottom 20% of the income distribution appeared to be more strongly associated with multimorbidity than those who have less than half the average income

(5)     there are too many tables--- 3 or 4 would be ideal...Tables 5-6 could be relegated to an appendix and discussed in text...table 6 with non-significant findings seems unnecessary

Answer: We agreed and moved Table 5-6 into Appendix. Accordingly, restated the result as below.

“…We investigated the effects of poverty stability on multimorbidity and the relationship between income changes and multimorbidity. Initial income appeared to be statistically significantly associated with multimorbidity. None of the income change measures were significant and presented the result in Appendix…

(6)     Finally and most importantly, I believe the conclusions section could be strengthened by discussing study limitations.....most notably I would discuss the absence of place-related and other quality of life/material well-being indicators or access to care/quality of care indicators. The study does not illuminate how/why poverty leads to multimorbidity. The possibility exists that it is not poverty per se but rather how policy influences residential adverse exposures, household material well-being, and access to quality care (as suggested in the introduction) and thus the exclusion of these other indicators results in strong impacts of poverty rather than the more proximal determinants of health. Similarly, there should be a discussion of income vs wealth in this context.

Answer: Thank you for your comment. We added limitation section There is no data available about place-related and other quality of life/material well-being indicators or access to care/quality of care indicators. We added those limitations on the discussion section and importance of further research.

Round 2

Reviewer 1 Report

Appreciate taking a good faith effort to revise based on my earlier comments. Findings of this revision, which differ from your first submitted version, shed more light on the scientific validity of your work. Good luck.

Author Response

Reviewer 1

Appreciate taking a good faith effort to revise based on my earlier comments. Findings of this revision, which differ from your first submitted version, shed more light on the scientific validity of your work. Good luck.

Answer: Thank you for your reviewing and good comment.
